# State-trait interactions in regulatory focus determine impulse buying behavior

**Anand Krishna**[ORCID]*, **Sophia Ried, Marie Meixner**

Department of Motivational and Emotional Psychology, Institute of Psychology, Julius-Maximilians-University, Würzburg, Bavaria, Germany

* krishna@psychologie.uni-wuerzburg.de

## Abstract

Little research has focused on motivational state-trait interactions to explain impulse buying. Although the trait chronic regulatory focus has been linked to impulse buying, no evidence yet exists for an effect of situational regulatory focus and no research has examined whether the fit of chronic and situational regulatory focus can influence impulse buying with actual consumptive consequences rather than purchase intentions. Two laboratory experiments (total $N = 250$) manipulated situational regulatory focus before providing opportunities for impulse buying. In addition, cognitive constraint was manipulated as a potential boundary condition for regulatory focus effects. Situational promotion focus increased impulse buying relative to situational prevention focus in participants with strong chronic promotion, consistent with regulatory fit theory and independently of cognitive constraint. Surprisingly, situational promotion focus also increased impulse buying in participants with strong chronic prevention, but only under low cognitive constraint. These results may be explained by diverging mediating cognitive processes for promotion vs. prevention focus' effect on impulse buying. Future research must focus more on combining relevant states and traits in predicting consumer behavior. Marketing implications are discussed.

## Introduction

The spontaneous urge to buy is something with which many human beings are intimately familiar. A full 68% of U.S. consumers reported having bought something on impulse in the last month in a 2012 survey [1], whereas in-store studies evaluating the extent of impulse buying behavior show that between 46% and 91% of products a consumer buys in a given trip may be impulsive purchases depending on the circumstances [2]. Given the large impact of impulse buying behavior on end-user consumption, much research has focused on discovering antecedents and predictors of impulse buying decisions. In the literature, *impulse buying* is usually described as a spontaneous act of purchasing with an associated positive emotional feeling, diminished regard for cost and a temptation to self-fulfil via consumption [3]. Although the timeframe for an impulsive purchase decision is often short, impulse buying urges may also persist over a longer period, albeit thereby increasing the likelihood that the impulse will be curbed by constraining cognitions [4, 5]. The majority of the work on impulse buying concerns direct

**Data Availability Statement:** All data, materials, and analysis script files are available from the OSF database (DOI: 10.17605/OSF.IO/4ZWEX).

**Funding:** This publication was supported by the Open Access Publication Fund of the University of Wuerzburg.

**Competing interests:** The authors have declared that no competing interests exist.

situational influences, such as the time allotted to the shopping trip [e.g. 2] or the degree of sensory stimulation in the store [6]. Another significant proportion focuses on trait-level predictors related to impulse buying both directly [e.g. 7] and indirectly [e.g. 8]. However, a recent meta-analysis [3] pointed to two important gaps in the literature: first, the authors report that the situational variables consisted mostly of external factors such as product type, environmental cues, available finances and social influence. Only few of the situational variables were motivational in nature: a hedonic or utilitarian goal of the purchase, involvement and importance of the purchase, and internal drive states. Taken together, these variables accounted for only 10.2% of the effects included. Given that motivation is a primary determinant of the direction and energization of human behavior [9], this dearth of studies should be addressed. The second gap identified by Amos and colleagues is the low number of studies that examine state-trait interactions. Only 8.7% of the effects they analyzed pertained to interactions between states and traits, but these effects were the greatest in magnitude [3]. As the interplay between stable motives and situational incentives has long been a focus of motivational psychology [e.g. 10], it seems appropriate to delve further into motivational determinants of impulse buying behavior.

A theory with particular potential for application to consumer behavior is *regulatory focus theory* [11]. The basic premise of regulatory focus theory is that human motivation may be divided into two relevant motivational states: a *promotion focus*, which is a motivational state concerned with nurturance goals and attaining positive outcomes, and a *prevention focus*, which is concerned with security goals and avoiding negative outcomes. In a promotion focus, individuals are likely to pursue goals in an eager manner, valuing positive outcomes more highly and thereby ignoring possible risks more readily, whereas a prevention focus will lead individuals to pursue goals vigilantly and in a risk-averse fashion due to their increased valuation of successful avoidance [12]. Importantly, a situational regulatory focus can be induced as a motivational state by appropriate goal framing, but it may also be measured as a disposition. Chronic regulatory focus is a trait measured separately for promotion and prevention focus that describes the person-specific likelihood to adopt a given focus in any situation. Individuals with a high chronic prevention focus are therefore likely to perceive goals in terms of avoidance of negative outcomes. For example, in a purchase decision, a chronically prevention-focused individual might generally tend to consider risks and negative outcomes of the purchase, such as the product's price or its durability, while a chronically promotion-focused individual might instead focus on the product's hedonic value or its potential to achieve a desired endstate [13].

Both situational and dispositional regulatory focus have been shown to affect consumer behavior. Prior research has demonstrated that consumers attend more to information that is relevant to goals consistent with their situational regulatory focus [14, 15], that these effects of regulatory focus may generalize to investment decisions [16–18], and that a promotion focus increases the likelihood that consumers will respond based on their spontaneous feelings [19]. A promotion focus has even been linked to increased (knowing) consumption of counterfeit goods [20]. Furthermore, effects of chronic regulatory focus have also been shown on consumer-relevant outcomes, such as the reception of advertisements [21, 22]. Importantly, multiple studies have built on this previous research and demonstrated that state-trait interactions or *regulatory fit* effects [23] can explain consumer behavior above and beyond the simple main effects or situational and dispositional regulatory focus [e.g. 24, 25]. For example, the fit of a hedonic vs. a utilitarian shopping context to a promotion vs. a prevention focus increases both consumer purchase intentions in and perceived value of e-commerce and m-commerce environments [26–28].

With regard to impulse buying specifically, Verplanken and Sato [29] provide a theoretical analysis in which they argue that impulse buying should be understood as a form of self-regulation. They conclude that although both a promotion and a prevention focus may lead to increased impulse buying behavior, a prevention focus is likely to do so only in the presence of

existing negative emotional states as a form of mood management. A promotion focus, on the other hand, has fewer requirements for facilitating impulse buying. The majority of promotion-focused strategies they identify are more broadly applicable to potential consumers. More recently, Das has applied the construct of chronic regulatory focus to predicting impulse buying behavior [8, 30]. His findings indicate that a chronic promotion focus is indeed associated with more impulsiveness and hedonic shopping values, factors which increase the likelihood of impulse buying, whereas a chronic prevention focus instead predicts lower impulsiveness, more utilitarian shopping values and greater store loyalty, factors that should lessen impulse buying [3]. These relationships between trait regulatory focus and impulse buying have been replicated in an online shopping context [31]. In addition, there is evidence that consumers are more motivated to impulse buy when the brand reflects their ideal self-image [32]. However, no extant research has experimentally examined the causal effect of regulatory focus or regulatory fit on actual impulse buying behavior.

## The current research

### Theoretical background

In order to fill this gap and gain a clearer understanding of how regulatory focus may influence impulse buying, the current research seeks to achieve two main goals: first, to demonstrate an effect of situational regulatory focus on actual consumption, and second, to explore regulatory fit effects that may impact its magnitude. A secondary goal of this research is to investigate boundary operating conditions for any effect of regulatory focus and/or fit in order to increase applicability to retailing contexts. In addition, though not the focus of this research, the design of the current studies allows for replication of some previous findings as well as exploratory analyses which are discussed in the supplementary analyses (S1 File).

In general, a promotion focus should lead to increased impulse buying behavior compared to a prevention focus. This prediction can be derived from basic regulatory focus theory due to an increased motivation to achieve hits (i.e. consume attractive goods) rather than correct rejections (i.e. avoid regret), at least in the absence of negative states which require active regulation [29, 33, 34]. As outlined earlier, existing research indirectly supports this prediction–chronic regulatory focus is associated with other variables that influence impulse buying [8, 30] and regulatory focus increases the impact of goal-congruent information [21, 22]. From the latter point, it can be derived that a promotion focus should increase impulse buying especially strongly when the consumer context suggests hedonic goals and the products are ready-to-use [35]. For this reason, the current studies implement a store selling various snacks and drinks, articles which are often bought for hedonic and instantaneous consumption [36]. Snacks and drinks are chosen to reflect a mix of less healthy (e.g. chocolate bars) and (relatively) healthier (e.g. fruit smoothies) options. Participants evaluate each article individually and decide whether to buy it relatively quickly, reflecting impulse buying behavior. In addition, articles in the store are discounted in order to foster a strong baseline of impulsive urges to buy [35]. In order to closely simulate actual buying behavior, participants can spend some or all of their compensation for taking part in the experiment on these products, mirroring the consequences for actual impulse buying behavior [37].

### Hypotheses

**H1:** Individuals in a situational promotion focus should spend more money when given the opportunity to impulsively buy than individuals in a situational prevention focus.

The prediction for chronic regulatory focus is similar, although it must be adjusted for the separate measures of chronic promotion and prevention focus. However, it merits its own

hypothesis, as it is unclear whether the predicted effects of chronic regulatory focus manifest in the presence of a strong situational induction, especially as prior research has found that chronic regulatory focus inclinations may be counteracted by such an induction [17].

**H2a:** Chronic promotion focus will be positively related to the amount of money spent on impulse buying.

**H2b:** Chronic prevention focus will be negatively related to the amount of money spent on impulse buying.

Regulatory fit effects occur when an aspect of the situation fits the regulatory focus of an individual. As this research focuses on state-trait interactions as predictors of impulse buying, regulatory fit is operationalized by having a strong chronic regulatory focus that fits the situational regulatory focus. As chronic regulatory focus can be measured separately for promotion and prevention, we consider regulatory fit effects likely to occur when an individual with a strong chronic promotion focus is placed in a situational promotion focus or when an individual with a strong prevention focus is placed in a situational prevention focus. Previous research suggests that regulatory fit should increase the effect of a given situational regulatory focus [23, 38], as reflected in findings in e-commerce where a chronic promotion focus increased purchase intentions in a hedonic shopping context, but a chronic prevention focus increased them in a utilitarian shopping context [26, 27]. However, the opposite pattern of fit effects has obtained in other research concerned specifically with impulsive behavior [17], in which the authors argued that the behavioral tendency induced by a strong chronic regulatory focus may be particularly vulnerable to an opposed situational regulatory focus. Thus, we make no directional prediction with regard to regulatory fit effects.

**RQ1:** Do situational and chronic regulatory focus interact to determine impulse buying?

Retail settings can contain many extraneous influences that can weaken the effect of motivational variables such as regulatory focus. For example, a lack of cognitive or opportunity resources may reduce deliberation about purchase decisions [5]. However, it is unclear whether regulatory focus effects on consumer behavior require such deliberation, as they can manifest independently of consumer involvement [22]. On the other hand, behavioral inhibition of impulses under time pressure has been demonstrated to be unaffected by regulatory focus [39]. Thus, the current study also investigates whether restricted cognitive resources moderate regulatory focus and fit effects on impulse buying.

**RQ2:** How does a lack of opportunity or capacity to deliberate impact the effects predicted in H1 and H2?

As noted above, the design of the current studies also allows for replication of existing work pertaining to the influence of time pressure on impulse buying [40] as well as the availability of self-regulatory resources during the purchase decision [41]. In addition, exploratory analyses can be conducted separately for healthier and less healthy products in order to test the generalizability of any findings pertaining to the research questions and hypotheses formulated above. These analyses are secondary to the main topic of this article, however, and will be discussed in the (S1 File).

## Experiment 1

### Method

The full materials and Inquisit 4.0 experimental scripts used for this experiment are available on the Open Science Framework (https://osf.io/tr3ga/?view_only=28166ecdb5214c8d a71446839929c5e9). Both this experiment and Experiment 2 were conducted in accordance

with the ethical guidelines of the Deutsche Gesellschaft für Psychologie (DGPs). As no negative impact on participants was expected that exceeded typical daily emotional experiences, no specific ethics approval was required under these guidelines. Informed consent was provided via signature of a detailed informed consent form, which was then countersigned by the experimenter.

**Participants and data collection.**  A total of 121 participants took part in the experiment. Three participants encountered technical errors during the experiment; their data was discarded. Of the remaining 118 participants (76.3% female, Age: $M$ = 27.2, $SD$ = 8.1), six refused to accept the products they had bought during the experiment and demanded the money, one was noted to have ignored all instructions by the experimenter and two did not speak fluent German. Data from these participants were discarded. In addition, one participant who had missed responding to more than one buying prompt in time was also excluded from analysis. The final sample consisted of 108 participants (77.8% female, Age: $M$ = 26.5, $SD$ = 7.1).

Data was collected at a German university and participants were recruited from the university mailing list (including predominantly students, but also a significant minority of employed and unemployed adults; ethnicity predominantly White Germans) as a convenience sample. The experiment took between 10 and 15 minutes and was situated in a longer session of one hour's total length. Participants were compensated with 7€ or whatever combination of goods and remaining budget with which they finished the experiment.

**Design and procedure.**  The design of the experiment was a 2x2 between-subjects design with the factors situational regulatory focus (promotion vs. prevention) and time pressure (low vs. high) and the additional covariate buying impulsiveness [7]. The final sample size was sensitive to an effect size of $f$ = .273 with power of at least 80%.

Participants were assessed in groups of up to six at once at individual computer workstations separated by dividers. After signing an informed consent form which was countersigned by the experimenter, participants completed the regulatory focus manipulation task. Next, participants completed the impulse buying measure. Thereafter, participants responded to several manipulation check items, followed by the measure of chronic regulatory focus, the measure of buying impulsiveness and a measure of eating behavior attitudes. Finally, participants provided demographic data, received their monetary payout and bought goods and were debriefed.

**Regulatory focus manipulation.**  The manipulation of situational regulatory focus was based on the mouse maze task [42], but adapted for computerized administration. The structure of the regulatory focus tasks was identical across the conditions, but the framing, target and signal stimuli, feedback presentation, and optimal strategy varied in order to address both the goal and means levels of regulatory focus [43].

All participants completed trials as follows: first, a cross was displayed in the middle of the screen. When participants clicked the cross, a picture of a mouse appeared in the middle of the screen and two target pictures (a mouse hole and a piece of cheese) appeared on either side of the mouse. In addition, a signal picture and an instruction text were displayed at the top of the screen. Participants were required to click the relevant target picture for their regulatory focus condition faster than a hidden time limit. The time limit started at 1000ms and adapted to participants' performance such that the current time limit was 1.14 times the median reaction time of the participant's last ten correct clicks. If participants did not click on any target within 1500ms, the trial ended. After each trial, participants received feedback on whether they had clicked the correct target and whether they had responded fast enough. Participants were instructed that their monetary reward depending on them achieving a specific performance criterion in this task.

In the promotion condition, the task was framed as leading the mouse to the cheese and the performance outcome was framed as a possible gain of 2€. Participants were instructed to lead the mouse to the cheese as fast as possible and that clicking correctly was not very important. The relevant target picture was the piece of cheese, the signal picture was a clear and sunny sky, and the signal text was "LAUF HIN" ("run to it"). The feedback text for response speed was presented in a larger font than the feedback text for response accuracy. The performance criterion was responding fast enough at least 80% of the time and correctly at least 60% of the time, implying an optimal strategy of favoring speed over accuracy.

In the prevention condition, the task was framed as saving the mouse from an owl and the performance outcome was framed as a possible loss of 2€. Participants were instructed to lead the mouse to the mouse hole as accurately as possible and that clicking correctly was very important. The relevant target picture was the mouse hole, the signal picture was that of a perched owl, and the signal text was "RENN WEG" ("run away"). The feedback text for response accuracy was presented in a larger font than the feedback text for response speed. The performance criterion was responding correctly at least 80% of the time and fast enough at least 60% of the time, implying an optimal strategy of favoring accuracy over speed.

Participants first completed four practice trials of this task after reading the initial instructions. They then completed four blocks of 40 trials each. After each block, participants were given performance feedback and reminded of the performance criterion. It should be noted that the actual monetary payout provided to the participants did not vary based on their performance, but was always displayed as 7€.

**Impulse buying measure.** After completing the regulatory focus task, participants were informed that they could use their compensation of 7€ from the regulatory focus manipulation task to buy various products. Participants saw pictures of various product groups of two to three related products from a common category (such as chocolate bars or smoothies, available from the OSF project) with an associated price (discounted by at least 30% compared to retail values) and were instructed to press the S key to buy one of the onscreen products and the L key to reject the onscreen products. In the low time pressure condition, participants had one minute per product to decide whether they wanted to buy, whereas in the high time pressure condition, participants had five seconds per product to decide. If the time limit for the decision elapsed, participants were informed that the computer would randomly decide whether to buy the product and that this decision would be binding for them. In both conditions, the time remaining for the decision was displayed on screen with each product group. After responding or allowing the time limit to elapse, participants saw feedback on the results of their response for one second (either that they had bought the product, that they could no longer buy the product because their budget was exhausted, or that they had not bought the product). If participants bought a product, the same product group was displayed again until it was rejected or the participant's budget was no longer sufficient to buy it. After all eight product categories had been completed, participants were shown a summary of the products they had bought and the money they had remaining. Product categories and associated prices are listed in the (S2 File), while pictures are available with the remaining materials on the OSF. Note that participants who bought a product (e.g. a soft drink) selected the specific type (e.g. Coca-Cola, Mezzo Mix or Fanta) after the experiment's completion.

**Manipulation checks and questionnaires.** Next, participants completed five manipulation check items. They were asked how important the 2€ reward for the mouse game had been to them, how important they considered accuracy during the mouse task, how important they considered speed during the mouse task (all five point Likert scales from 'not at all important' to 'very important'), how they had understood the goal of the mouse task (five point Likert scale from 'protecting mouse from harm' to 'getting rewards for mouse'), and to what degree

they felt time pressure during the purchase decisions (five point Likert scale from 'no time pressure at all' to 'strong time pressure'). Thereafter, they completed the measure of chronic regulatory focus [44, 45], the measure of buying impulsiveness [7], and the Dutch Eating Behavior Questionnaire [46], which will not be further discussed in this paper due to its lack of direct relevance.

## Results

Raw data and analysis scripts are available on the Open Science Framework (https://osf.io/ 489mj/?view_only=88905a940d6c4dd8902ba80e3319489d). Full descriptive data and model estimates are provided in the methodological data appendix (S3 File). All analyses were performed with IBM SPSS v.24.

**Scale reliabilities and descriptive data.** The BIS achieved excellent internal consistency ($\alpha$ = .910), while the chronic promotion focus scale achieved acceptable reliability ($\alpha$ = .737) and the chronic prevention focus scale achieved poor reliability ($\alpha$ = .655). On average, participants indicated that the reward in the mouse game was of strong importance to them ($M$ = 4.15, $SD$ = .97). Of the entire sample, 55.6% spent money on products voluntarily (over entire sample: $M$ = 1.22€, $SD$ = 1.86€). Additional descriptive statistics may be found in the methodological appendix (S3 File).

**Manipulation checks.** Participants indicated that they had construed the task more as getting rewards for the mouse in the promotion focus condition ($M$ = 4.52, $SD$ = .81) compared to the prevention focus condition ($M$ = 3.27, $SD$ = 1.54), $t$(75.99) = 5.23, $p$ < .001, $d$ = 1.03. However, participants did not indicate the expected preferences for speed or accuracy (all $ts$ < 1.3, all $ds$ < .24). For the time pressure manipulation, participants in the high time pressure condition ($M$ = 3.29, $SD$ = 1.31) reported feeling more time pressure than those in the low time pressure condition ($M$ = 2.19, $SD$ = 1.37), $t$(106) = 4.26, $p$ < .001, $d$ = .82. This effect was corroborated by observed reaction time differences (high time pressure: $M$ = 1371, $SD$ = 732; low time pressure: $M$ = 3417, $SD$ = 1992), $t$(65.35) = 7.03, $p$ < .001, $d$ = 1.35.

**Data analysis.** For the analysis, all metric predictors were grand-mean centered. The dependent variable for all analyses was money spent voluntarily. Money spent by the computer's random decision when a participant did not respond in time was not included. As the effect of interest was the effect of regulatory focus on impulse buying independent of existing propensities towards impulse buying, the BIS scale was included as a covariate in analyses. A 2x2 between-subjects ANCOVA with the factors situational regulatory focus (promotion vs. prevention) and time pressure (low vs. high) and the covariate BIS yielded a significant main effect of situational regulatory focus ($F$(1,103) = 7.52, $p$ = .007, $\eta_p^2$ = .068). No other effects achieved significance (all $Fs$ < 1, all $ps$ > .34, all $\eta_p^2$ < .01). Participants in a situational promotion focus spent more money ($M$ = 1.72€, $SD$ = 2.07€) than participants in a situational prevention focus ($M$ = .69€, $SD$ = 1.44€), in line with $H1$.

To evaluate $H2$ and $RQ2$, a regression model was calculated that included the metric predictors chronic promotion focus, chronic prevention focus, time pressure condition as a dummy variable (0 = low, 1 = high), the interaction terms between both chronic regulatory focus measures and the time pressure condition, as well as BIS as a control variable. This approach allowed us to control for BIS in an analogous fashion to the ANCOVA analysis. No predictors achieved significance (all $|Bs|$ < .76, all $ps$ > .15). However, simple bivariate correlations between the metric predictors and money spent showed a positive correlation between chronic promotion focus and money spent (see Table 1). These results provide partial, but weak support for $H2$. In addition, lack of opportunity to deliberate did not moderate the effect of either situational or chronic regulatory focus, providing the first indications towards resolving $RQ2$.

**Table 1. Correlations of metric predictors with money spent in Experiment 1.**

|  | BIS | Chronic promotion focus | Chronic prevention focus |
|---|---|---|---|
| Money spent | .138 | .191* | -.083 |
| BIS | - | .089 | -.182† |
| Chronic promotion focus |  | - | .133 |
| Chronic prevention focus |  |  | - |

*Note.* *: $p \leq .05$,
†: $p \leq .10$.

To investigate *RQ1*, the state-trait interaction of promotion and prevention focus were examined separately to limit the number of predictors in the regression models, as the sample size was slightly below Green's recommended level for testing multiple individual predictors [47]. Therefore, two separate regression models were calculated that each included: either chronic promotion or prevention focus score as a main predictor, time pressure condition (0 = low, 1 = high) and situational regulatory focus (0 = promotion, 1 = prevention) as dummy variables, and all interaction terms between these variables. In addition, BIS and the respective other chronic regulatory focus score were included as control variables, the latter as recommended by Shah and colleagues [48].

In the promotion focus model, the predictors that achieved significance were situational regulatory focus ($B = -1.12$, $SE = .50$, $p = .022$) and chronic promotion focus ($B = 1.65$, $SE = .63$, $p = .011$), although this was qualified by a marginally significant chronic promotion state-trait interaction ($B = -1.94$, $SE = 1.10$, $p = .080$). No other predictors achieved significance (all | $Bs| < .89$, all $ps > .18$). This pattern of results is illustrated in Fig 1. For participants one standard deviation below the average chronic promotion focus, there was no significant effect of situational regulatory focus ($B = -.21$, $SE = .75$, $p = .785$), whereas for participants one standard

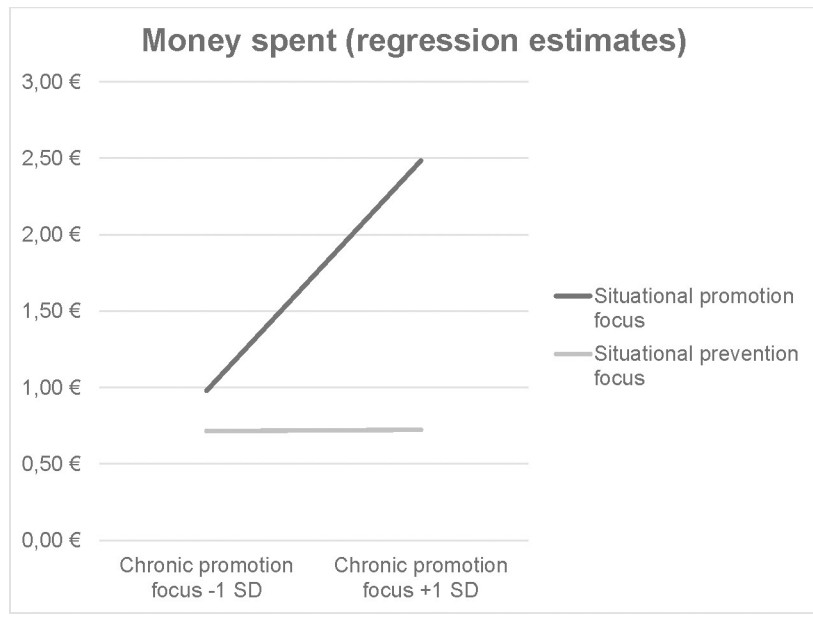

**Fig 1. Interaction of chronic and situational regulatory focus based on regression estimates from the full promotion focus model, Experiment 1.**

deviation above the average chronic promotion focus, this effect obtained ($B$ = -2.15, $SE$ = .74, $p$ = .005).

The prevention focus model was structured similarly to the promotion focus model, only substituting chronic prevention focus for chronic promotion focus. However, the only predictor in the model that achieved significance was situational regulatory focus ($B$ = -1.28, $SE$ = .51, $p$ = .014; all other $|Bs| < .1.69$, all other $ps > .11$). Descriptively, the data trended towards a positive effect of chronic prevention focus on impulsive spending in the situational promotion focus group, but a negative effect in the situational prevention focus group. This pattern of results suggests that situational regulatory focus can indeed impact impulse buying behavior, but that its effect may be moderated by the consumer's chronic regulatory focus.

## Discussion

Experiment 1 provided initial evidence that individuals are influenced by regulatory focus in their impulse buying behavior. Participants in a situational promotion focus spent more than twice the amount that participants in a situational prevention focus spent. Furthermore, this effect was not impacted by time pressure during the purchase decision, indicating that regulatory focus effects can generalize to fast decisions. Although chronic regulatory focus was not directly related to impulse buying behavior, exploratory analyses suggested that it may play a moderating role in determining the impact of situational regulatory focus.

However, there are some weaknesses of Experiment 1 that must be addressed. First, although there was an effect of situational regulatory focus on the main dependent measure, several manipulation check variables did not show the expected effect, casting some doubt on what was actually achieved by the manipulation. Participants did not report favoring speed more or accuracy less in the promotion focus condition, although they did show a strong framing effect of protecting the mouse vs. rewarding the mouse. Second, although participants reported feeling more time pressure in the high time pressure group, the actual mean decision time in both groups was well below the five second time limit in the high time pressure group. Therefore, this experiment provides little insight whether regulatory focus effects unfold via relatively resource-dependent cognitive operations, as the purchase decisions in general seem to require little time. Third, the statistical evidence in support of the hypotheses and conclusions drawn was fairly weak in this experiment. The interaction term of situational regulatory focus and chronic promotion focus was not significant by a standard criterion and there was no converse effect for chronic prevention focus. This may be partially attributable to the relatively small sample size of Experiment 1. With cell sizes of between 25 and 28 participants in the final sample, the power of this study to detect effects might have been too low to properly investigate a state-trait interaction effect. For these reasons, a second experiment was conducted to replicate the results and address these problems.

## Experiment 2

### Method

The full materials and Inquisit 4.0 experimental scripts used for this experiment are available on the Open Science Framework (https://osf.io/szmcq/?view_only=96e1f435654f4393813679 89f1d2bf4a). The experiment was also preregistered on the Open Science Framework.

**Participants and data collection.** A total of 201 participants took part in the experiment. Two participants did not provide complete data due to technical reasons and were excluded from analysis. Of the remaining 199 participants (72.4% female, Age: $M$ = 26.4, $SD$ = 9.0), in accordance with the pre-registration, participants were excluded for anomalies in the log (zero participants), for indicating they did not believe that the purchase decisions were real at the

time they made them (53 participants), for failing to make a purchase decision within the time limit more than once (one participant), for ignoring the alphanumeric load strings from the cognitive load manipulation in more than half of the purchase decisions (two participants), for indicating they were not at all conscientious in memorizing the alphanumeric load strings (zero participants), and for making serious errors as defined in the preregistration script in at least half of the load trials (one participant). The final sample therefore consisted of 142 participants (71.1% female, Age: $M = 25.5$, $SD = 7.4$).

Data collection proceeded under similar circumstances to Experiment 1 at the same university. Participants were compensated with 5€ or whatever combination of goods and remaining budget with which they finished the experiment.

**Design and procedure.** The design of the experiment was identical to Experiment 1, except that the factor time pressure was replaced by cognitive load (low vs. high). The final sample size was sensitive to an effect size of $f = .237$ with power of at least 80%.

The procedure of Experiment 2 was identical to Experiment 1 except where noted.

**Regulatory focus manipulation.** The mouse task from Experiment 1 was adapted to increase its power to affect the tactical level of regulatory focus [43]. Participants again completed trials in which they symbolically led a mouse to certain targets. As before, trials began with participants clicking a cross in the middle of the screen. After clicking, the cross was immediately replaced by a picture of a mouse and two identical target stimuli appeared on either side of the screen. After 500ms, an antagonist appeared at the top of the screen and moved to either side of the screen over the next 300ms. Participants were required to move the mouse cursor to the target stimuli which was free of the antagonist before the antagonist completed the movement (responses were possible before the antagonist appeared). After each trial, participants' score changed depending on whether they had moved to the correct target and whether they had responded fast enough. Participants were instructed that their monetary reward depending on them achieving a specific performance criterion in this task.

In the promotion condition, the task was framed as leading the mouse to cheese and the performance outcome was framed as a possible gain of 2€. Both target stimuli were pictures of cheese. The antagonist was another mouse who was framed as taking some of the participants' cheese. Participants were instructed to gain cheese points. They gained three cheese points by moving fast enough and one by moving to the correct target, implying an optimal strategy of favoring speed over accuracy. The size of the feedback text emphasized the speed outcome and the color of the points feedback changed between red and green depending on this outcome.

In the prevention condition, the task was framed as saving the mouse from an owl and the performance outcome was framed as a possible loss of 2€. Both target stimuli were pictures of mouse holes. The antagonist was an owl who was framed as threatening the mouse. Participants were instructed to avoid losing mouse lives. They lost three mouse lives by moving to the wrong target and one by moving too slowly, implying an optimal strategy of favoring accuracy over speed. The size of the feedback text emphasized the accuracy outcome and the color of the lives feedback changed between red and green depending on this outcome.

Participants completed one block of 40 trials. The performance criterion in the promotion task was achieving at least 90 cheese points, whereas in the prevention condition it was not losing more than 70 mouse lives, ensuring that participants who followed the optimal strategy (speed or accuracy) always succeeded (expected gain: 120 cheese points vs. expected loss: 40 mouse lives). It should be noted that the actual monetary payout provided to the participants was always 5€, regardless of their performance.

**Impulse buying measure.** The impulse buying measure was the same as in Experiment 1, except that all participants had a time limit of 15 seconds per purchase decision. Instead of varying time pressure, a cognitive load manipulation was introduced. Before each purchase

decision, participants saw a randomly generated alphanumeric load string for five seconds. They were instructed to remember these alphanumeric strings during their purchase decision. After buying as many of the following product category as they wanted (or could afford), participants were asked to reproduce the alphanumeric string. If they reproduced it correctly, the screen showed "CORRECT" for one second before the next string was shown. If they reproduced it incorrectly, the screen showed error feedback for two seconds before the next string was shown. In the low cognitive load condition, the alphanumeric strings consisted of two characters (e.g. "a2"). In the high load condition, the strings consisted of seven characters (e.g. "a7ksm21").

**Manipulation checks and questionnaires.**   Participants were again asked how important the 2€ reward for the mouse game had been to them. The manipulation check items from Experiment 1 on speed and accuracy were merged into a single item asking what aim participants mostly followed during the mouse task (five point Likert scale from 'being exact, avoiding the wrong side' to 'being fast, not missing any chances'), how they had understood the goal of the mouse task (five point Likert scale from 'protecting mouse from harm' to 'getting rewards for mouse'), and how they perceived the reward outcome in the mouse task (five point Likert scale from 'not losing money' to 'earning additional money'). They were further asked how difficult remembering the alphanumeric strings was (five point Likert scale from 'not at all difficult' to 'very difficult'), how much remembering the strings impaired their purchase decisions, and how well they could concentrate on the purchase decisions (five point Likert scale from 'not at all' to 'extremely'). In addition, they were asked how conscientious they were in remembering the alphanumeric strings (five point Likert scale from 'not at all' to 'extremely'), whether they stopped trying to remember the strings after a certain point and if so, at which product. Finally, they were asked whether they made the purchase decisions in the experiment in the truthful expectation of actually receiving the products and having to pay for them with their compensation money and if not, why not. The chronic regulatory focus and BIS measures were the same as in Experiment 1, but the Dutch Eating Behavior Questionnaire was omitted.

## Results

Raw data and analysis scripts are available on the Open Science Framework (https://osf.io/f4xut/?view_only=cec40384baa7467fb0d2ce5975126a23).

**Scale reliabilities, descriptive data and manipulation checks.**   The BIS again achieved excellent internal consistency ($\alpha = .926$), while both the chronic promotion focus ($\alpha = .777$) and chronic prevention focus ($\alpha = .713$) scales achieved acceptable reliability. On average, participants indicated that the reward in the mouse game was of importance to them ($M = 3.98$, $SD = 1.15$). Of the entire sample, 57.0% spent money on products voluntarily (over entire sample: $M = 0.89€$, $SD = 1.12€$). Table 2 shows the results of manipulation checks for the manipulation of situational regulatory focus, while Table 3 shows the manipulation checks for the cognitive load manipulation. All measures differed strongly between conditions in the expected direction. Additional descriptives may be found in the methodological appendix (S3 File).

**Preregistered data analysis.**   In accordance with the preregistered analysis script, data was prepared and analyzed in the same way as Experiment 1 except where noted. A 2x2 between-subjects ANCOVA with the factors situational regulatory focus (promotion vs. prevention) and cognitive load (low vs. high) and the covariate BIS yielded no significant effects (all $Fs < 2.2$, all $ps > .14$, all $\eta_p^2 < .02$). *H1* failed to replicate ($F = .287$, $p = .593$, $\eta_p^2 = .002$).

*H2* was evaluated with separate regression models for promotion and prevention focus to increase comparability with the subsequent state-trait interaction analysis. Each model

**Table 2. Results of manipulation checks for situational regulatory focus in Experiment 2.**

| | Promotion focus (n = 72) | Prevention focus (n = 70) | t(140) | Cohen's d |
|---|---|---|---|---|
| | M (SD) | M (SD) | | |
| Framing performance outcome as earning extra money | 3.94 (1.21) | 2.03 (1.26) | 9.24 | 1.55 |
| Understanding goal as earning rewards for mouse | 4.28 (1.12) | 2.39 (1.47) | 8.66 | 1.45 |
| Favoring speed over accuracy | 3.99 (1.27) | 2.47 (1.45) | 6.62 | 1.11 |
| Average reaction time in mouse game | 476ms (287) | 807ms (221) | 7.69 | 1.29 |
| Accuracy in mouse game | 54.5% (10.9) | 81.2% (17.5) | 10.92 | 1.83 |

*Note.* All self-report items measured on Likert scales from 1 to 5. All measures differed between groups at $p < .001$.

included the relevant chronic regulatory focus, cognitive load condition as a dummy variable (0 = low, 1 = high) and their interaction term. The other chronic regulatory focus and the BIS score were included as control variables. No individual predictors in the promotion model achieved significance (all $|Bs| < .28$, all $ps > .14$). The same held for the prevention model (all $|Bs| < .36$, all $ps > .14$). Simple correlations (non-preregistered) also show no relationship between chronic regulatory focus and impulse buying (see Table 4). These results provide no support for *H2*. In addition, cognitive load did not moderate the effect of either situational or chronic regulatory focus.

As in Experiment 1, two further regression models were calculated to investigate state-trait interactions. No individual predictors achieved significance in the promotion focus model (all $|Bs| < .32$, all $ps > .21$). However, two predictors in the prevention focus model did achieve significance: the chronic prevention state-trait interaction ($B = -1.23$, $SE = .50$, $p = .015$) and the three-way interaction of chronic prevention focus, situational regulatory focus and cognitive load ($B = 1.36$, $SE = .69$, $p = .049$). This pattern of results is shown in Fig 2. A situational promotion focus increases spending most strongly for participants with a strong chronic prevention focus who also have low cognitive load ($B = -.69$, $SE = .44$, $p = .124$). In all other trait-load combinations, the situational regulatory focus effect is weaker or reversed (all $|Bs| < .75$, all $ps > .104$).

## Discussion

Although Experiment 2 had a larger sample size than Experiment 1 and showed strong effects on all manipulation checks, the main effect of situational regulatory focus failed to replicate. However, the state-trait analyses once again produced some evidence that the effects of regulatory focus on impulse buying may be moderated by individuals' chronic dispositions. Unlike in Experiment 1, where chronic promotion focus determined the effect of the situational regulatory focus manipulation, in this experiment, chronic prevention focus appeared to play a greater role. In addition, the cognitive load manipulation moderated the influence of chronic prevention focus on the situational regulatory focus effect. The weak statistical evidence makes

**Table 3. Results of manipulation checks for cognitive load in Experiment 2.**

| | Low Load (n = 70) | High Load (n = 72) | t(140) | Cohen's d |
|---|---|---|---|---|
| | M (SD) | M (SD) | | |
| Ability to concentrate on purchase decisions | 4.01 (.94) | 3.06 (1.21) | 5.26 | .88 |
| Difficulty reproducing strings after purchase decision | 1.37 (.64) | 3.65 (1.16) | 14.40 | 2.42 |
| Distraction from purchase decision by remembering strings | 1.37 (.78) | 2.76 (1.42) | 7.21 | 1.21 |

*Note.* All items measured on Likert scales from 1 to 5. All measures differed between groups at $p < .001$.

**Table 4. Correlations of metric predictors with money spent in Experiment 2.**

|  | BIS | Chronic promotion focus | Chronic prevention focus |
|---|---|---|---|
| Money spent | .016 | -.014 | .104 |
| BIS | - | .221*** | -.113 |
| Chronic promotion focus | | - | -.107 |
| Chronic prevention focus | | | - |

*Note.* ***: $p \leq .001$,

*: $p \leq .05$,

†: $p \leq .10$.

drawing conclusions difficult, however. For this reason, the data from both studies are subjected to a mega-analysis in order to increase the power of the analyses to discriminate between spurious and robust effects.

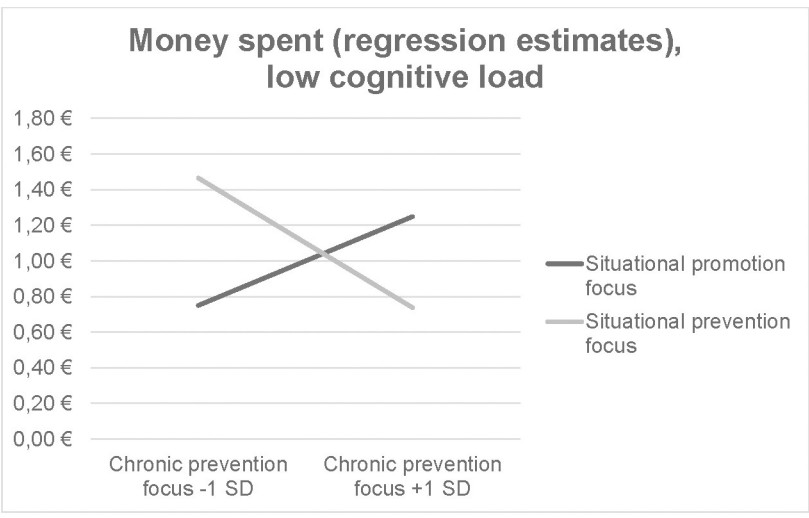

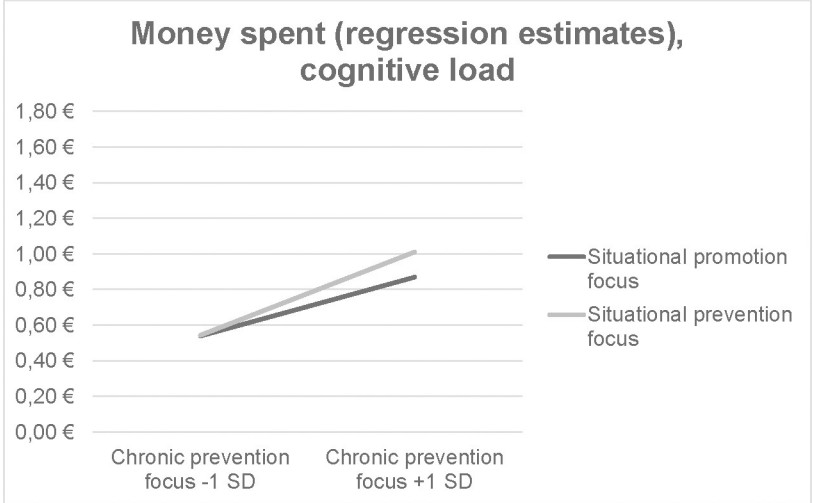

**Fig 2. Interaction of chronic and situational regulatory focus with cognitive load based on regression estimates from the full prevention focus model, Experiment 2.**

## Mega-analysis

The final samples from both experiments were combined. In order to make the dependent measures comparable across the experiments, the amount of money spent was converted to a percentage of budget spent. The percentage of budget spent did not differ across the studies, $t(248) = .083$, $p = .934$, $d = .011$. The final sample consisted of 250 participants (74.0% female, Age: $M = 25.9$, $SD = 7.3$). Of the entire sample, 56.4% spent money on products voluntarily (over entire sample: $M = 17.6\%$, $SD = 24.3$). Additional descriptive statistics may be found in the methodological appendix (S3 File).

A 2x2 between-subjects ANCOVA with the factors situational regulatory focus (promotion vs. prevention) and cognitive resource constraint (low, i.e. low time pressure/low cognitive load) vs. high, i.e. high time pressure/high cognitive load) and the covariate BIS yielded a marginally significant main effect of situational regulatory focus ($F(1,245) = 2.82$, $p = .094$, $\eta_p^2 = .011$) and no other effects (all $Fs < 1.7$, all $ps > .20$, all $\eta_p^2 < .01$). If there is a simple effect of situational regulatory focus in the population, it is of small magnitude based on this sample.

Similarly to the individual experiments, two regression models were calculated to investigate state-trait interactions. In the promotion focus model, the predictor chronic promotion focus achieved significance ($B = .12$, $SE = .06$, $p = .043$) and the chronic promotion state-trait interaction achieved marginal significance ($B = -.16$, $SE = .09$, $p = .057$; all other $|Bs| < .07$, all $ps > .14$). This pattern is shown in Fig 3. Individuals with a low chronic promotion focus show no effect of situational regulatory focus ($B = .02$, $SE = .06$, $p = .780$), but those with a high chronic promotion focus do show such an effect ($B = -.15$, $SE = .06$, $p = .017$). As shown in Experiment 1, manipulations of situational regulatory focus impact the impulse buying of chronically promotion-focused individuals more than that of those with a weak chronic promotion focus.

The prevention focus model shows a significant main effect of chronic prevention focus ($B = .12$, $SE = .05$, $p = .028$) which is qualified by a significant chronic prevention state-trait interaction ($B = -.26$, $SE = .08$, $p = .002$) and a three-way interaction of the state-trait interaction with the resource constraint condition ($B = .24$, $SE = .12$, $p = .044$). The pattern is similar to

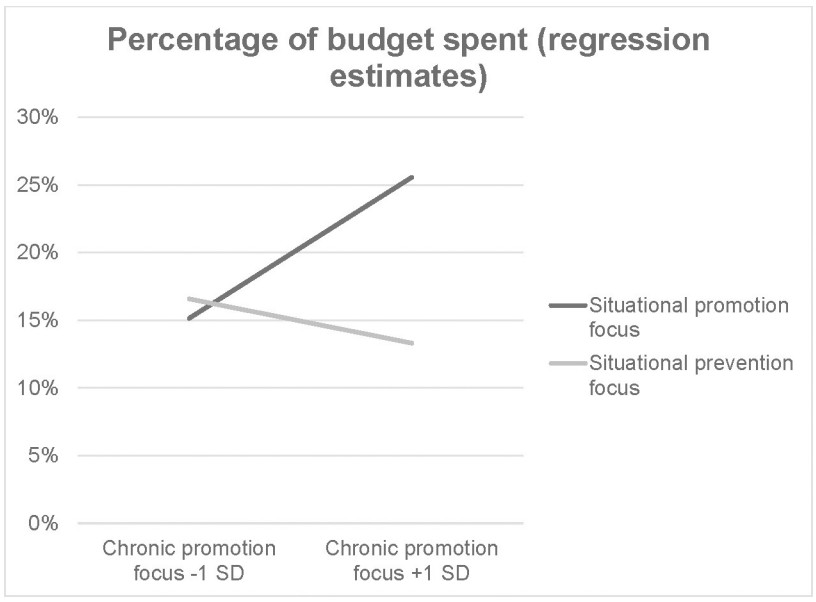

**Fig 3. Interaction of chronic and situational regulatory focus based on regression estimates from the full promotion focus model, mega-analysis.**

Experiment 2 and can be seen in Fig 4. The only significant effect of situational regulatory focus occurs when participants have a strong chronic prevention focus and unconstrained resources ($B = -.19$, $SE = .06$, $p = .002$), all other combinations show no effect of situational regulatory focus (all $|Bs| < .08$, all $ps > .27$). Chronically prevention-focused buyers are impacted by a situational regulatory focus cue only if they have resources to elaborate the decision.

To summarize, situational regulatory focus only impacts impulse buying behavior when participants have a chronic promotion or a chronic prevention focus. When participants are predisposed to either of these motivational orientations, a situational promotion focus increases impulse buying behavior, whereas a situational prevention focus decreases it. Interestingly, a chronic promotion focus appears to increase the impact of a situationally fitting promotion focus regardless of the availability of cognitive resources, but a chronic prevention focus only increases the effect of situational regulatory focus when cognitive resources are available. This indicates that the latter effect might be driven by more elaborative reasoning than the presumably more efficient promotion focus effect. In line with these results, simple bivariate correlations show no effects of chronic regulatory focus on money spent (see Table 5); this effect only

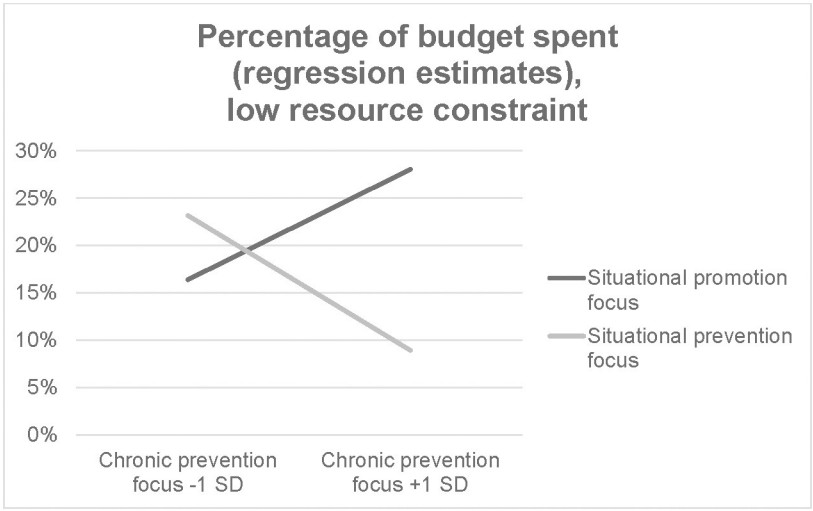

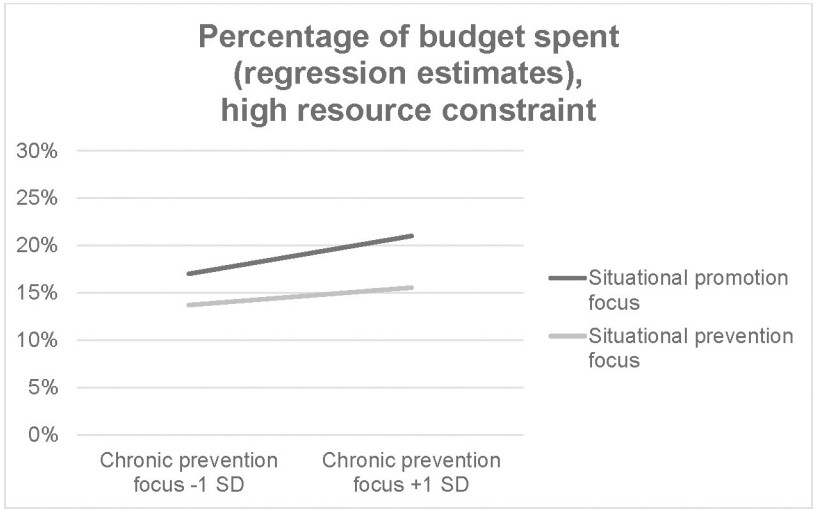

**Fig 4. Interaction of chronic and situational regulatory focus with resource constraint based on regression estimates from the full prevention focus model, mega-analysis.**

**Table 5. Correlations of metric predictors with money spent, mega-analysis.**

|  | BIS | Chronic promotion focus | Chronic prevention focus |
|---|---|---|---|
| Money spent | .092 | .076 | .024 |
| BIS | - | .198*** | -.178*** |
| Chronic promotion focus |  | - | -.010 |
| Chronic prevention focus |  |  | - |

*Note.* ***: $p \leq .001$.

manifests in combination with situational factors. However, chronic promotion and prevention focus do correlate with BIS scores in a manner consistent with past research [8, 30, 31].

## General discussion

This research set out to investigate whether the motivational state of a promotion focus increases actual impulse buying behavior relative to a state prevention focus and whether individuals' chronic disposition towards a given regulatory focus impacts impulse buying directly or via modulation of state regulatory focus. The results of two experiments taken individually were mixed, with the first providing evidence for a general effect of state regulatory focus and statistically weaker evidence for a moderating effect of chronic promotion focus and the second not replicating these effects. A mega-analysis of the experiments provided qualified support for a promotion fit increasing impulse buying behavior. The secondary question of this research investigated whether cognitive resource availability influences regulatory focus and fit effects on impulse buying behavior. The results of the mega-analysis showed that resource availability constrains the effect of prevention, but not of promotion fit, with prevention fit effects occurring only when resources were available for elaboration.

### Operational and theoretical implications

While the two experiments reported here were similar in design, they produced different results. It is unlikely that design variations are responsible for these differences, however. The major differences between the experiments concern the manipulations of regulatory focus and cognitive constraint as well as the participants' budget for impulse buying. Although the manipulation of regulatory focus in the first experiment did not achieve significance on all manipulation checks, the task framing measure did show a significant effect in line with past conceptualizations of regulatory focus manipulations [42]. Furthermore, this experiment produced a main effect of situational regulatory focus on money spent. Conversely, the second experiment showed strong effects of the manipulation on all manipulation checks. It is thus unlikely that the manipulation failed to induce a situational regulatory focus in either experiment. The cognitive constraint manipulation differed chiefly in the requirement for a response between individual purchase decisions. In Experiment 1, participants made impulse buying decisions sequentially without interruption, possibly establishing a sense of flow that might have been weakened in Experiment 2, where buying decisions were punctuated by entry of the cognitive load strings. As a state of flow is a potential mechanism for regulatory fit effects [49], this may have attenuated such effects in Experiment 2, but it is unclear why flow effects should have disappeared entirely in a predictable task such as this. Finally, participants could spend 7€ in Experiment 1, but only 5€ in Experiment 2. However, participants spent similar percentages of their budget across both experiments, rendering an influence of this design factor unlikely.

These experiments produced complex and contradictory results, but would thus seem comparable in design. Therefore, it is the results that remain stable in the mega-analysis that should be interpreted. Surprisingly, neither situational nor chronic regulatory focus affected impulse buying on their own. Instead, the analysis indicated that a situational promotion focus increased impulse buying both for chronically promotion-focused participants and for chronically prevention-focused participants, but for the latter only when they were free of cognitive constraint. This may indicate that different processes underlie the influence of chronic promotion and prevention focus on impulse buying. A promotion focus has been shown to favor consumption decisions based on spontaneous feelings [19], whereas a prevention focus generally increases elaborative reasoning [22, 50]. Thus, the promotion fit effects in these experiments might be driven more by fast and cognitively efficient processes, such as a lowered threshold for behavioral activation, while prevention fit effects may require more effortful elaboration to manifest in a consumer context, for example by biasing reasoned decision-making processes. A further possibility suggested by the finding that promotion fit effects were weaker in Experiment 2 is that these specifically were driven by flow in the purchase decisions, which may be been weakened in that experiment as mentioned above. Future research might address the precise cognitive processes involved.

In the context of existing findings connecting regulatory focus with impulse buying [8, 30, 31], the current studies extend and elaborate this research. Correlations between chronic measures of regulatory focus and trait impulse buying proclivities were replicated, but the experimental approach shows that effects on actual money spent are contingent on activation of a fitting situational regulatory focus. Although future research must address whether a strong chronic focus increases the impact *only* of the corresponding situational focus (as suggested by the promotion fit results), whether it increases *general* sensitivity to situational regulatory focus cues (as partially suggested by the prevention fit results), or whether both possibilities occur simultaneously mediated by separate processes, the current results suggest that future regulatory focus research on impulse buying should take regulatory fit into account. Thus, these conclusions dovetail with Amos and colleagues' [3] recommendation to investigate state-trait interactions as determinants of impulse buying rather than only state or trait variables individually.

Additional results of this research also shed some light on other findings from the impulse buying literature. As discussed in the (S1 File), the results of additional analyses indicate that time pressure or lack of cognitive resources in themselves do not affect impulse buying in the absence of an actual retail environment, presenting a possible boundary condition for previous findings [51, 52]. In addition, the existing finding that depletion of self-regulatory resources increases the impact of buying impulsiveness traits on actual impulse spending [41] did not extend to circumstances where cognitive resources are bound by another task, even though these resources are assumed to have a common basis [53]. This may indicate that there is something special about the sequential task paradigm used in the depletion studies that the current research could not capture, or it may point towards the conclusion that depletion effects are weaker than previously assumed [54, 55]. It seems that there is still room to more closely examine the role of time and cognitive constraint on impulse buying behavior, especially with regard to mediating processes and whether the constraint applies only to the purchase target itself (such as when travel websites imply that the current offer is only available for a very short time) or to the general shopping situation (such as when one must finish a full shopping trip before one becomes late for another appointment).

Finally, exploratory analyses differentiating between healthy and unhealthy products (suggested by an anonymous reviewer) presented some intriguing possibilities for future research (see S1 File for a full report). As unhealthy foods present both potentially greater rewards (due

to their likely better taste, [56]) and greater risks (due to their adverse health consequences in addition to the potential waste of money), they should lead to stronger regulatory focus effects. Indeed, the promotion fit effect did appear to be slightly stronger for unhealthy products compared to healthy products. However, the prevention fit effect manifested for healthy products independently of cognitive constraint, but for unhealthy products only in participants free of cognitive constraint. To the degree that the prevention fit effect was indeed driven by relatively resource-intensive biased decision-making processes as discussed above, these findings suggest that the presence of multiple risk cues (unhealthiness *and* wasting money) may increase their costs so much that they can no longer impact behavior in the presence of cognitive constraints. However, these post-hoc analyses are not sufficiently robust to draw strong inferences. Future research might investigate these hypotheses to shed more light on the link between regulatory focus and impulsive behavior.

## Marketing implications

This research replicated the extant finding that a chronic promotion focus is associated with a stronger and a chronic prevention focus with a weaker predisposition towards impulse buying. As different cultures are associated with different chronic regulatory foci [28, 57], one obvious implication for marketing is that providing impulse buying opportunities is more likely to reap dividends in developed countries (which tend to be promotion-oriented) than developing countries (which tend to be prevention-oriented). Other, more specific cultural values such as collectivism or individualism can also predict chronic regulatory focus, allowing for more fine-grained application of this recommendation to specific countries [58]. However, the unique contribution of these experiments lies in the potential regulatory fit effects they illuminate.

While these effects are inconsistent in parts, one pattern reoccurred throughout: a situational promotion focus can increase impulse buying in individuals with either a strong chronic promotion or a strong chronic prevention focus. Although retailers do not have control over the traits of their customers, they can adapt their retail environment to foster a specific regulatory focus, for example by tailoring their advertising messages to be either gain- or loss-focused [59, 60]. Investing resources to induce a situational promotion focus for potential impulse buyers may be most profitable in environments where strong chronic foci predominate. Counter-intuitively, the current findings suggest that a situational promotion focus might even increase impulse buying for chronically prevention-oriented individuals, although this appears only to hold when there is no cognitive constraint. Therefore, retailers looking to maximize impulse buying in prevention-oriented customer bases might be well-advised to avoid tactics such as sales countdowns, which may increase perceived time pressure [61].

It should finally be noted that the prevailing wisdom on regulatory fit effects, namely that they enhance the effect of the dominant focus [25], is not fully borne out in this research. Although the results are partially in line with this expectation, the finding that a chronic prevention focus can increase the effect of a situational promotion focus under some circumstances in particular is surprising. It may be that impulsive behavior (including impulse buying) is specifically likely to produce such paradoxical effects, in line with past findings [17]. Another important difference to previous studies is that the consumption *goal* in these studies was arguably always hedonic rather than utilitarian. Thus, regulatory fit effects based on the consumption goal [e.g. 15] might not be applicable to our paradigm. Instead, our findings highlight that risks and potential gains associated with the products themselves may interact with consumer predispositions to determine impulsive behavior. However, this is an open question for future research to address. Marketers should be wary of assuming that regulatory fit effects will always favor the dominant focus.

A limitation of this study for application to marketing are the relatively low budgets and costs involved for the consumers. Many impulse purchases are made in more potentially costly product categories such as clothing. It is unclear whether our effects generalize to such contexts. However, the use of a paradigm with actual spending consequences ameliorates this issue somewhat [37]. Participants invested a significant proportion of their compensation for the experiment (on average 10–15% of the total per product they bought). Although this was a small sum in absolute terms, it would likely have felt like a fairly large cost in comparison to the reference point–the difference between going home with 2€ instead of 5€. Therefore, the application of our findings to low-cost, low-consequence impulse buying behavior seems adequate. Our findings also suggest that a situational promotion focus can increase spending even in participants who are dispositionally likely to focus on costs (i.e. high chronic prevention focus participants). Thus, it seems plausible that even higher-cost impulse buying decisions may be affected similarly.

Finally, our studies used only foodstuffs as the products. As one-use products with a mainly hedonic use, this product category represents comparatively simple consumption decisions in which involvement is likely low. Thus, generalizing to more complex products or decision contexts is difficult. However, our exploratory analyses suggest tentatively that more complex goods (i.e. those associated with potential risks in consumption beyond monetary expenditure) may require marketers to allow consumers to process the decision more thoroughly in order for any situational regulatory focus cues to show an effect, as otherwise chronic prevention-focused consumers in particular might remain indifferent to such cues.

## Conclusion

This research attempts to address the surprising lack of research focusing on motivational states in the impulse buying literature. It provides evidence that a situational promotion focus can increase impulse spending in a low-cost low-consequence environment relative to a situational prevention focus, particularly in individuals with strong chronic regulatory focus dispositions. For chronically promotion-focused individuals, this effect is strengthened independently of cognitive constraints, but for chronically prevention-focused individuals, it is only strengthened when individuals are cognitively unconstrained. Both basic and applied research may benefit from considering the interaction of dispositions and states when predicting behavior.

## Supporting information

**S1 File. Analyses concerning conceptual replication of findings unrelated to the main paper's focus and exploratory analyses concerning product type.**
(PDF)

**S2 File. Prices and wares available during the impulse buying task.**
(PDF)

**S3 File. Full model estimates and descriptive data for all analyses.**
(PDF)

## Author Contributions

**Conceptualization:** Anand Krishna, Sophia Ried.

**Data curation:** Anand Krishna, Marie Meixner.

**Formal analysis:** Anand Krishna.

**Methodology:** Anand Krishna.

**Project administration:** Anand Krishna, Sophia Ried, Marie Meixner.

**Software:** Anand Krishna, Sophia Ried, Marie Meixner.

**Supervision:** Anand Krishna.

**Writing – original draft:** Anand Krishna.

**Writing – review & editing:** Anand Krishna, Sophia Ried, Marie Meixner.

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
