## [Decision Letter · Decision Letter 0]

8 Apr 2021

PONE-D-20-26264

State-trait interactions in regulatory focus determine impulse buying behavior

PLOS ONE

Dear Dr. Krishna,

Thank you for submitting your manuscript to PLOS ONE. After careful consideration, we feel that it has merit but does not fully meet PLOS ONE’s publication criteria as it currently stands. Therefore, we invite you to submit a revised version of the manuscript that addresses the points raised during the review process.

We look forward to receiving your revised manuscript.

Kind regards,

Luigi Cembalo, PhD

Academic Editor

PLOS ONE

Journal Requirements:

Your ethics statement should only appear in the Methods section of your manuscript. If your ethics statement is written in any section besides the Methods, please move it to the Methods section and delete it from any other section. Please ensure that your ethics statement is included in your manuscript, as the ethics statement entered into the online submission form will not be published alongside your manuscript.

Reviewers' comments:

Reviewer's Responses to Questions

**Comments to the Author**

1. Is the manuscript technically sound, and do the data support the conclusions?

Reviewer #1: Partly

Reviewer #2: Yes

2. Has the statistical analysis been performed appropriately and rigorously? 

Reviewer #1: Yes

Reviewer #2: N/A

3. Have the authors made all data underlying the findings in their manuscript fully available?

Reviewer #1: Yes

Reviewer #2: Yes

4. Is the manuscript presented in an intelligible fashion and written in standard English?

Reviewer #1: No

Reviewer #2: Yes

5. Review Comments to the Author

Reviewer #1: The manuscript seeks to demonstrate the effect of situational regulatory focus on actual consumption and to explore the impact of regulatory fit effects. A secondary goal of this research is to investigate boundary operating conditions for any effect of regulatory focus and/or fit in order to increase applicability to retailing contexts.

It would be very important to further deepen on the marketing implications that may be derived from the analysis . The conclusions should be drawn more appropriately based on the data presented.

The English language could be improved in several points, thus a rereading is strongly suggested.

Reviewer #2: The manuscript deals with a study on motivational state-trait interactions to explain impulsive buying. In particular, to gain a clearer understanding of how regulatory focus may influence impulse buying, the study seeks to achieve two main goals: first, to demonstrate an effect of situational regulatory focus on actual consumption, and second, to explore regulatory fit effects that may impact its magnitude. As emphasized by the authors, a secondary goal of this research is to investigate boundary operating conditions for any effect of regulatory focus and/or fit in order to increase applicability to retailing contexts. The study is interesting and original. The authors clearly identify the gaps that exist in the literature on this topic.

However, there are some aspects that need to be clarified in the revision phase.

1) Experimental design. The authors in the two experiments used ready-to-use products which are bought for hedonic and instantaneous consumption. It is right for the purpose of the study, but an accurate description of the set of products used in the experiments are needed. More in particular, in experiment 1 you wrote (page 12 lines 258-259): "Participants saw pictures of various product groups of two to three related products from a common category (such as chocolate bars or smoothies)". I think a more accurate description of this phase is necessary. I suggest providing one or two pictures shown during the experiments.

In addition, did you find differences in the responses of the participants that could be related to the different types of products (eg. chocolate bars or smoothies)? For example, the literature on food consumption revealed that consumers may react differently on the basis of the type of product they may buy. One might imagine, for example, that a smoothie could be perceived as healthier than a chocolate bar and therefore impulsively more pleasing. Anyway, I suggest giving more emphasis to the products used in the experiments.

2) Data analyses. To evaluate the H2, the authors wrote (page 15 line 317) that a regression model was calculated. But which regression model did you use? The same in line 330.

3) Discussion of results, implications and conclusion. In these parts, it is also important to refer to the category of products used in the experiments, as the results can probably not be generalized to all product categories, nor to complex goods.

6. PLOS authors have the option to publish the peer review history of their article (what does this mean?). If published, this will include your full peer review and any attached files.

Reviewer #1: No

Reviewer #2: No

---

## [Author Response · Author response to Decision Letter 0]

10 May 2021

The following is a copy of the cover letter:

Dear Professor Cembalo,

We thank you for the opportunity to revise our manuscript “State-trait interactions in regulatory focus determine impulse buying behavior”. We have addressed each of the reviewers’ points and believe the manuscript has been strengthened as a result. We have also adapted the manuscript to conform to the PLOS ONE style requirements, including moving the ethics statement. Please find our point-by-point responses to the reviewers below. For ease of reading, our responses are presented in italics.

Reviewer #1: The manuscript seeks to demonstrate the effect of situational regulatory focus on actual consumption and to explore the impact of regulatory fit effects. A secondary goal of this research is to investigate boundary operating conditions for any effect of regulatory focus and/or fit in order to increase applicability to retailing contexts.

It would be very important to further deepen on the marketing implications that may be derived from the analysis . The conclusions should be drawn more appropriately based on the data presented.

We have substantially reworked our Marketing Implications section to include more limitations of our design and qualify our conclusions. Specifically, we have added some clarification distinguishing our study from previous work (l. 715ff), as well as a discussion of the effect of the relatively low costs for products and of the product types emphasizing the assumptions needed to generalize our findings (ls. 722-742). 

The English language could be improved in several points, thus a rereading is strongly suggested.

We have reread the entire text and made some stylistic improvements at various points. The first author is a native English speaker with a UK background and has carefully reviewed the manuscript’s language; unfortunately, we are unsure what exactly the reviewer is criticizing. We hope we have addressed this concern.

Reviewer #2: The manuscript deals with a study on motivational state-trait interactions to explain impulsive buying. In particular, to gain a clearer understanding of how regulatory focus may influence impulse buying, the study seeks to achieve two main goals: first, to demonstrate an effect of situational regulatory focus on actual consumption, and second, to explore regulatory fit effects that may impact its magnitude. As emphasized by the authors, a secondary goal of this research is to investigate boundary operating conditions for any effect of regulatory focus and/or fit in order to increase applicability to retailing contexts. The study is interesting and original. The authors clearly identify the gaps that exist in the literature on this topic.

We thank the reviewer for these words of encouragement.

However, there are some aspects that need to be clarified in the revision phase.

1) Experimental design. The authors in the two experiments used ready-to-use products which are bought for hedonic and instantaneous consumption. It is right for the purpose of the study, but an accurate description of the set of products used in the experiments are needed. More in particular, in experiment 1 you wrote (page 12 lines 258-259): "Participants saw pictures of various product groups of two to three related products from a common category (such as chocolate bars or smoothies)". I think a more accurate description of this phase is necessary. I suggest providing one or two pictures shown during the experiments.

We agree that the design of our store and the purchase decisions in particular was not clear. We have added a figure (Fig 1) showing example stimuli and added a sentence explaining how participants selected their final product: “Note that participants who bought a product (e.g. a soft drink) selected the specific type (e.g. Coca-Cola, Mezzo Mix or Fanta) after the experiment’s completion.” (l. 282ff)

In addition, we hope that interested readers will access our experimental files on the OSF, where all materials and the experiment software are available. We have also uploaded the analysis code for the mega-analysis.

In addition, did you find differences in the responses of the participants that could be related to the different types of products (eg. chocolate bars or smoothies)? For example, the literature on food consumption revealed that consumers may react differently on the basis of the type of product they may buy. One might imagine, for example, that a smoothie could be perceived as healthier than a chocolate bar and therefore impulsively more pleasing. Anyway, I suggest giving more emphasis to the products used in the experiments.

We thank the reviewer for this suggestion! The resulting analyses differentiating between healthy and unhealthy foodstuffs produced interesting results, although we interpret them with caution due to their post-hoc nature. We have added some text at several points in the manuscript noting this differentiation:

“Snacks and drinks are chosen to reflect a mix of less healthy (e.g. chocolate bars) and (relatively) healthier (e.g. fruit smoothies) options.” (l. 133ff)

“In addition, exploratory analyses can be conducted separately for healthier and less healthy products in order to test the generalizability of any findings pertaining to the research questions and hypotheses formulated above.” (ls. 182-185)”

Analyses split by product type have been added to the supplementary materials (S1 Supplementary Analyses); in the interest of brevity, we decided against including these in the main text. We discuss their implications from ls. 673-687 and refer to them again in the marketing implications (ls. 735-742).

2) Data analyses. To evaluate the H2, the authors wrote (page 15 line 317) that a regression model was calculated. But which regression model did you use? The same in line 330.

The initial formulation of these sentences was obtuse, as the model description was not presented straight away; we have modified the descriptions to be easier to read:

“To evaluate H2 and RQ2, a regression model was calculated that included the metric predictors chronic promotion focus, chronic prevention focus, time pressure condition as a dummy variable (0 = low, 1 = high), the interaction terms between both chronic regulatory focus measures and the time pressure condition, as well as BIS as a control variable. This approach allowed us to control for BIS in an analogous fashion to the ANCOVA analysis.” (ls. 333-337)

“To investigate RQ1, the state-trait interaction of promotion and prevention focus were examined separately to limit the number of predictors in the regression models, as the sample size was slightly below Green’s recommended level for testing multiple individual predictors (47). Therefore, two separate regression models were calculated that each included: either chronic promotion or prevention focus score as a main predictor, time pressure condition (0 = low, 1 = high) and situational regulatory focus (0 = promotion, 1 = prevention) as dummy variables, and all interaction terms between these variables. In addition, BIS and the respective other chronic regulatory focus score were included as control variables, the latter as recommended by Shah and colleagues (48). “ (ls. 345-353)

3) Discussion of results, implications and conclusion. In these parts, it is also important to refer to the category of products used in the experiments, as the results can probably not be generalized to all product categories, nor to complex goods.

As mentioned in the response to point 1, we now discuss the role of the product types in more detail (ls. 673-687 and 735-742).

Please address all correspondence regarding this submission to Anand Krishna, University of Würzburg, e-mail krishna@psychologie.uni-wuerzburg.de. 

 Yours sincerely,

 Anand Krishna, Sophia Ried & Marie Meixner

---

## [Decision Letter · Decision Letter 1]

10 Jun 2021

State-trait interactions in regulatory focus determine impulse buying behavior

PONE-D-20-26264R1

Dear Dr. Krishna,

We’re pleased to inform you that your manuscript has been judged scientifically suitable for publication and will be formally accepted for publication once it meets all outstanding technical requirements.

Kind regards,

Luigi Cembalo, PhD

Academic Editor

PLOS ONE

Additional Editor Comments (optional):

I went trough the document and I see you have improved your manuscript even beyond reviewers' requirements. I fully agree that you manuscript is ready for publication in PLOS-ONE as it is.

Reviewers' comments:

Reviewer's Responses to Questions

**Comments to the Author**

1. If the authors have adequately addressed your comments raised in a previous round of review and you feel that this manuscript is now acceptable for publication, you may indicate that here to bypass the “Comments to the Author” section, enter your conflict of interest statement in the “Confidential to Editor” section, and submit your "Accept" recommendation.

Reviewer #1: All comments have been addressed

Reviewer #2: All comments have been addressed

2. Is the manuscript technically sound, and do the data support the conclusions?

Reviewer #1: Yes

Reviewer #2: Yes

3. Has the statistical analysis been performed appropriately and rigorously? 

Reviewer #1: Yes

Reviewer #2: N/A

4. Have the authors made all data underlying the findings in their manuscript fully available?

Reviewer #1: Yes

Reviewer #2: Yes

5. Is the manuscript presented in an intelligible fashion and written in standard English?

Reviewer #1: Yes

Reviewer #2: Yes

6. Review Comments to the Author

Reviewer #1: (No Response)

Reviewer #2: The manuscript is much improved, all my comments have been addressed. I have no further suggestions.

7. PLOS authors have the option to publish the peer review history of their article (what does this mean?). If published, this will include your full peer review and any attached files.

Reviewer #1: No

Reviewer #2: No

---

## [Editor Report · Acceptance letter]

24 Jun 2021

PONE-D-20-26264R1 

State-trait interactions in regulatory focus determine impulse buying behavior 

Dear Dr. Krishna:

I'm pleased to inform you that your manuscript has been deemed suitable for publication in PLOS ONE. Congratulations! Your manuscript is now with our production department. 

Kind regards, 

on behalf of

Dr. Luigi Cembalo 

Academic Editor

PLOS ONE